# Automated Paraphrase Quality Assessment Using Language Models and Transfer Learning

**Bogdan Nicula** [1,*], **Mihai Dascalu** [1,2] , **Natalie N. Newton** [3], **Ellen Orcutt** [4] and **Danielle S. McNamara** [3]

1   Department of Computer Science, University Politehnica of Bucharest, 313 Splaiul Independentei, 060042 Bucharest, Romania; mihai.dascalu@upb.ro
2   Academy of Romanian Scientists, Str. Ilfov, Nr. 3, 050044 Bucharest, Romania
3   Department of Psychology, Arizona State University, P.O. Box 871104, Tempe, AZ 85287, USA; nnnewton@asu.edu (N.N.N.); dsmcnama@asu.edu (D.S.M.)
4   Department of Educational Psychology, University of Minnesota, 56 East River Road, Minneapolis, MN 55455, USA; orcut039@umn.edu
*   Correspondence: bogdan.nicula@upb.ro; Tel.: +40-720-284-840

**Abstract:** Learning to paraphrase supports both writing ability and reading comprehension, particularly for less skilled learners. As such, educational tools that integrate automated evaluations of paraphrases can be used to provide timely feedback to enhance learner paraphrasing skills more efficiently and effectively. Paraphrase identification is a popular NLP classification task that involves establishing whether two sentences share a similar meaning. Paraphrase quality assessment is a slightly more complex task, in which pairs of sentences are evaluated in-depth across multiple dimensions. In this study, we focus on four dimensions: lexical, syntactical, semantic, and overall quality. Our study introduces and evaluates various machine learning models using handcrafted features combined with Extra Trees, Siamese neural networks using BiLSTM RNNs, and pretrained BERT-based models, together with transfer learning from a larger general paraphrase corpus, to estimate the quality of paraphrases across the four dimensions. Two datasets are considered for the tasks involving paraphrase quality: ULPC (User Language Paraphrase Corpus) containing 1998 paraphrases and a smaller dataset with 115 paraphrases based on children's inputs. The paraphrase identification dataset used for the transfer learning task is the MSRP dataset (Microsoft Research Paraphrase Corpus) containing 5801 paraphrases. On the ULPC dataset, our BERT model improves upon the previous baseline by at least 0.1 in F1-score across the four dimensions. When using fine-tuning from ULPC for the children dataset, both the BERT and Siamese neural network models improve upon their original scores by at least 0.11 F1-score. The results of these experiments suggest that transfer learning using generic paraphrase identification datasets can be successful, while at the same time obtaining comparable results in fewer epochs.

**Keywords:** paraphrase quality assessment; natural language processing; recurrent neural networks; language models; transfer learning

## 1. Introduction

Paraphrases range widely in terms of definitions, from concise text constructs that are "similar enough in meaning" [1] to more philosophical implications, as paraphrases provide "differing textual realizations of the same meaning" [2]. In general, a paraphrase is a restatement of a text generated with different words, normally with the aim of providing clarity. Within an educational setting, the ability to paraphrase becomes vital, especially for young learners. Encouraging readers to transform a source text into more familiar words and phrases helps them better understand the text by activating relevant prior knowledge, as they develop a textbase model of what was explicitly conveyed in the text [3]. Learning to paraphrase facilitates both reading comprehension and writing ability, particularly for less skilled readers and writers [4–6]. An inability to generate a paraphrase is a clear

indicator that the reader is struggling with comprehension [7]. Moreover, learning how to effectively paraphrase provides a crucial foundation for students to master other skills that enhance reading comprehension, such as bridging and elaboration [4].

From a computational perspective, techniques that enable machines to automatically discriminate or generate paraphrases afford useful steps toward solving multiple Natural Language Processing (NLP) [8] problems. In past decades, techniques to identify paraphrases have been used to recognize redundancies and enable better text summarization, improve text generation strategies for language generating systems, or increase recall for information retrieval engines [9]. More recently, automated paraphrase generation was still used in the context of Natural Language generation for increasing the diversity of the generated text [10].

In the field of NLP, paraphrase identification is a popular task that involves assessing whether a pair of sentences constitutes a paraphrase. The task is usually modeled as a binary classification task (e.g., the Microsoft Paraphrase Research Corpus (MSRP) [1] offers only two labels—paraphrase or non-paraphrase), but it can vary depending on the dataset [11] (e.g., the Semantic Textual Similarity task from SemEval (STS) [12] rates sentence pairs on a scale from 0—sentences cover different topics—to 5—the sentences are completely equivalent). Paraphrase quality assessment is a related task, in which the pairs of sentences are scored across multiple dimensions. This approach captures a more complete perspective regarding the similarities and differences between two sentences. Depending on the number of dimensions and the granularity of the scoring, however, paraphrase quality assessment requires more effort on data generation.

Because of the simplicity of the task, paraphrase identification is a considerably more popular problem in the NLP community when compared to paraphrase quality assessment. The corpora used for paraphrase identification range from medium-sized datasets, such as MSRP with 5801 sentence pairs, to large datasets, such as the Quora Question Pair dataset (QQP) [13] with 400,000 paraphrase pairs. Given the large difference between the sizes of the datasets, a wide variety of machine learning models have been employed. For medium-sized datasets, a successful approach was to extract a set of handcrafted features, such as overlap features or latent sentence-level features, and train a supervised classification algorithm on top of these features [14]. Other approaches avoid manually defining the sets of features, and instead aggregate word embeddings through different pooling operations [15]. By contrast, deep learning NLP models are preferred for large datasets. In the case of QQP, one of the common elements that most top performing models share is the usage of a Bidirectional Encoder Representations from Transformers (BERT) encoder [16]. These models can be introduced alongside innovative masking techniques [17], or leverage smaller representations such as TinyBERT [18]. However, not all top performing models are based on BERT as there are deep neural networks using GLOVE embeddings [19] that obtain comparable results, with better runtime and less memory usage [20].

Despite the differences in style and content quality, both types of datasets share one shortcoming: they provide very little information regarding the quality of the paraphrase, as they solely indicate whether a given pair of sentences are a paraphrase or not. To our knowledge, the sole dataset that includes rubric scores regarding quality is the User Language Paraphrase Corpus (ULPC) [21], which scores paraphrases on 10 aspects using a point range from 1 to 6 and is described in detail in the Method section.

## 2. Current Study Objective

Our overarching objective is to implement scoring methods for paraphrases which can be used to develop a feedback system for a new version of iSTART (Interactive Strategy Training for Active Reading and Thinking [22]), called iSTART-Early, for young, developing readers (ages 9–11). Within iSTART, students improve their grasp on comprehension strategies (e.g., elaboration, bridging), by reading and self-explaining texts guided by

adaptive instructions. The end goal of iSTART is to improve students' comprehension of challenging texts and increase their performance in science courses.

This study introduces and evaluates various machine learning models, including handcrafted features combined with Extra Trees, Siamese neural networks using BiLSTM RNNs, and pretrained BERT-based models, together with transfer learning from a larger general paraphrase corpus, to estimate the quality of paraphrases across the four dimensions: lexical, syntactical, semantic, and overall quality. Predicting each of the dimensions for a given pair of sentences was considered as a separate classification task. We also assess the generalization capabilities of the previous models when building a specific model tailored for a small paraphrase corpus based on children inputs.

### 3. Method

#### *3.1. Corpus*

This study considers three different datasets (see Table 1): (a) the Microsoft Research Paraphrase Corpus (MSRP), (b) the User Language Paraphrase Corpus (ULPC), and (c) a small dataset containing paraphrases generated by children [23]. The datasets are focused either on paraphrase identification (i.e., the generic task in which pairs of sentences are labeled as being paraphrases or not) or on paraphrase quality assessment (i.e., a more complex task in which the quality of a paraphrase is assessed across multiple dimensions).

**Table 1.** Table describing the 3 datasets used as part of this study.

| Name | Number of Samples | Train/Test Split | Dimensions |
|---|---|---|---|
| MSRP | 5801 | 4076/1725 | 1 |
| ULPC | 1998 | 1012/649 | 4 |
| Children | 115 | 80/35 | 4 |

The MSRP is a paraphrase identification dataset containing 5801 paraphrase pairs (4076 for training and 1725 for testing). Each pair was extracted from news sources and was manually labeled with '1' if it is a paraphrase or '0' if it is not a paraphrase; the distribution of the two classes is 2:1.

The ULPC dataset contains 1998 paraphrase pairs annotated by trained human readers and divided into training (1012), validation (337), and test (649) datasets. The raters completed 50 h of training on a separate dataset prior to annotating ULPC entries. The sentence pairs were scored on a scale from 1 to 6 for four distinct paraphrase dimensions: semantic similarity, syntactic similarity, lexical similarity, and paraphrase quality. Because of the small number of samples in the dataset, the four dimensions were transformed into categories to reduce the complexity of the task; as such, binary (1.00–3.49 versus 3.5–6.00), or tripartite assessments (1.00–2.66, 2.67–4.33, and 4.33–6.00 only for the paraphrase quality dimension) were used.

The children dataset consists of only 115 paraphrase pairs extracted from responses provided by third and fourth graders. They were a group of 13 English Language learners who took part in a summer school program. Their text productions were corrected for obvious spelling errors prior to human scoring. The sentence pairs were scored on the same four paraphrase dimensions from ULPC, and were categorized into binary evaluations (i.e., for lexical, syntactic and semantic similarities) and tripartite evaluations (for paraphrase quality).

The annotations for both the children and the ULPC datasets were done in pairs of two raters. Where disagreement between the two raters occurred, a second evaluation step between the raters occurred before reaching a final, agreed upon result. Agreement between raters is presented in Table 2. Kappa's main advantage is that it corrects for chance agreement. Typical Kappa evaluations are for nominal categories; however, the ratings in this task are at the interval level. As such, both linear or a quadratic weighting schemes were employed to ensure that differences between ratings are accounted for (e.g., 1 and 3 are judged as more similar than ratings of 1 and 5). The low Kappa scores for paraphrase

quality on both datasets and for syntactic similarity on ULPC show that the raters disagreed more when scoring these dimensions. This indicates that these dimensions might be more difficult to assess by a human rater, and inherently by an automated model trained on the results generated by the human raters. Paraphrase quality, for instance, was especially difficult to score initially for the children dataset when the paraphrase strongly resembled a sentence that was copied word-for-word (i.e., copy-pasted). After discussions between the two raters, it was determined that modifying one word in the paraphrase was not sufficient for a paraphrase quality score above 0 (the lowest score).

**Table 2.** Kappa evaluations for Lexical (Lex), Syntactical (Syn), Semantical (Sem) similarity, and Paraphrase quality (PQ).

| Kappa | Lex | Syn | Sem | PQ |
|---|---|---|---|---|
| ULPC weighted Linear | 0.45 | 0.25 | 0.56 | 0.28 |
| ULPC weighted Quadratic | 0.62 | 0.43 | 0.71 | 0.43 |
| Children | 0.62 | 0.61 | 0.54 | 0.36 |

The ULPC and children datasets were preprocessed, eliminating poor sentence-paraphrase pairs that manifested one of the following four issues:

- garbage input—the response consists of random letters and has no meaning;
- irrelevant input—the response is blatantly irrelevant to the source sentence;
- shortness—one word responses were eliminated; and
- copy-paste input—responses that overlapped completely with the source, while adding no new words, were removed.

The filtering was made either based on annotated information (for garbage and irrelevancy issues) or on simple rule-based filters (for shortness and copy-paste issues).

High differences between the ULPC and children datasets are observed when considering the relations between the three rubric dimensions (semantic, syntactic, and lexical similarity) and paraphrase quality (see Figure 1). For the ULPC dataset, the mean value for lexical and semantic similarity increases with the paraphrase quality, while the syntactic similarity does not appear to be correlated with it. In contrast, the same relations for lexical and semantic similarity in the children dataset were not observed, as their means seem to decrease as paraphrase quality improves, although all three show strong correlations to paraphrase quality.

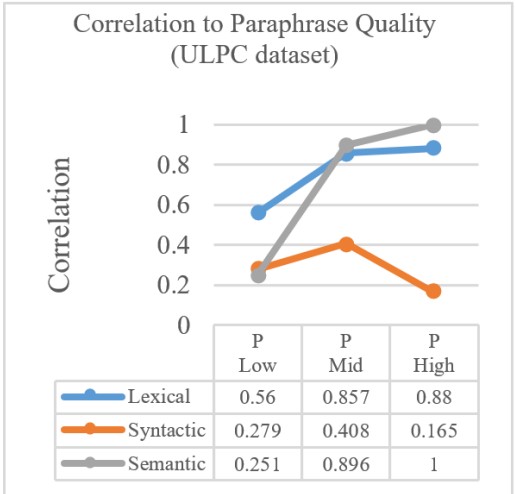 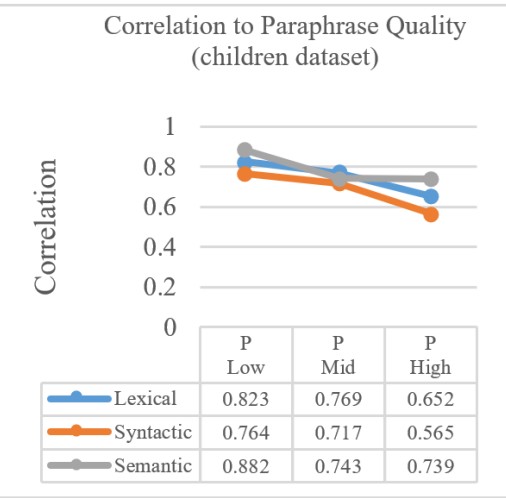

**Figure 1.** Correlations between paraphrase quality and the three rubric scores: semantic similarity, lexical similarity, and syntactic similarity for the children and ULPC datasets.

### 3.2. Models for Predicting Paraphrase Quality

Three different models were considered for predicting paraphrase quality, all described in detail in the following sub-sections. The first model relies on classical machine learning models and feature engineering. The model computes handcrafted features on top of which a machine learning classifier is used to predict different dimensions of paraphrase quality. The second model relies on Bidirectional Long Short-Term Memory (BiLSTM [24]) layers and a Siamese network (SN) architecture [25]. The third model consists of a pretrained BERT-based model fine-tuned for the current tasks.

The previous models were selected based on their complementary traits, each model providing different advantages, with corresponding shortcomings. The first model requires more effort in the initial stages for finding the best set of features, but it is fast to train and should manage well on small datasets. However, it has a major disadvantage, namely, that it does not support fine-tuning or transfer learning; instead, the model needs to be retrained for a new scenario. The Siamese neural network and the BERT-based models do not require any data preprocessing; nevertheless, their training is more resource-intensive and can require an exploration of the hyperparameter space to find the optimal learning rate. The SN model has fewer parameters than BERT architecture; thus, it should be easier to train and should be less prone to overfitting on small datasets. BERT-based models yield state-of-the-art performance on a wide array of NLP tasks, but overfitting can easily occur when working with small datasets.

### 3.2.1. Feature Engineering

Our first approach relies on extracting a wide array of features, filtering them, and using a classical machine learning classifier to make the predictions. Several types of features that evaluate the differences and similarities between two input sentences were used:

1. Levenshtein distance [26] between the source and paraphrase, while considering words as units or elements.
2. Overlap indices for both words and part-of-speech (POS) tokens (e.g., the percentage of words in the paraphrase that also appear in the source text).
3. Complexity indices related to surface, lexical, syntactic, and semantic properties computed using the ReaderBench framework [27] separately for the two sentences, as well as the absolute difference between them.
4. Complexity indices related to text cohesion computed in-between the two input sentences.

In total, 2368 features were generated. These features underwent a filtering stage, eliminating constant and highly correlated values, resulting in 594 features. Those features were used as input to train several machine learning classifiers from the SciKit Learn library [28] to predict each targeted dimensions. The following four models were evaluated: two variants of Support Vector Classifier (SVC–linear kernel and SVC–RBF kernel), Extra Trees (ET), and Multi-Layered Perceptron (MLP). Out of these models, the ET model consistently obtained the best results. In order to further reduce the feature space, a second filtering of the features was performed. The top 100 most predictive features of a trained Extra Trees model were selected, and an Extra Trees model was trained using only these features.

### 3.2.2. Siamese Neural Network

The second model involved a Siamese network architecture to extract and refine in parallel features from both sentences, then combine their deep representations to make a prediction. BiLSTM layers were used as building blocks for this architecture, as shown in Figure 2. Pretrained 300-dimensional GloVe [19] or word2vec [29] word embeddings were used, while the words that did not appear in the dictionary index were matched to a common 'unknown' embedding. Results are reported for the model using the GloVe embeddings, which surpassed in all configurations the alternative that employed word2vec embeddings.

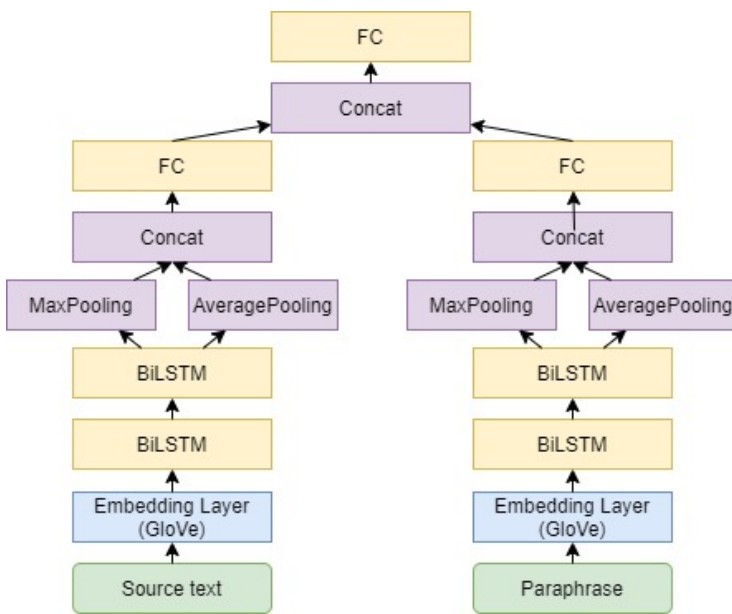

**Figure 2.** Siamese architecture using as a first input the source text and as a secondary input the paraphrase.

### 3.2.3. BERT-Based Neural Architecture

The third model relies on a pretrained BERT-based model from the Huggingface library [30]. For simplicity, a small version of BERT was used (identified as "bert-base-cased"). This model is composed of 12 layers comprising 110 million parameters and was trained on English texts from the BookCorpus and English Wikipedia. The two input sentences are passed to the model as a single string delimited by a special separator. The output of the BERT model goes into a Dropout layer, followed by a fully connected (FC) layer, which makes the final 2-class or 3-class predictions, as shown in Figure 3. In case of transfer learning experiments, this final layer was replaced by a random layer, keeping the remainder of the ensemble unchanged. Different learning rates were considered for BERT and FC components in order to help the model learn new patterns, while keeping as much as possible of the initial BERT pretrained knowledge.

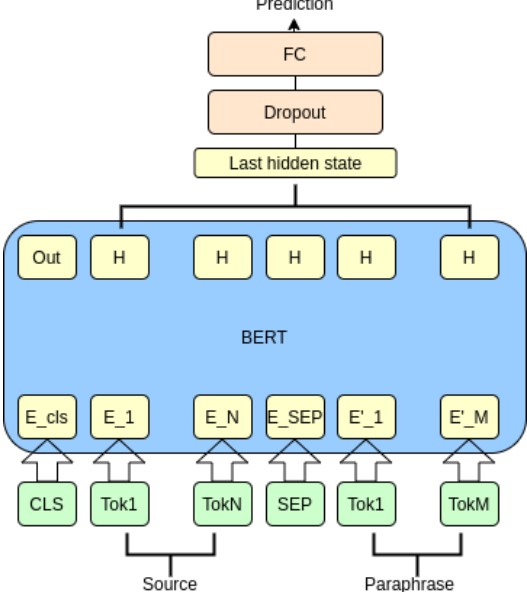

**Figure 3.** BERT-based architecture.

### 3.3. Fine-Tuning and Transfer Learning

Fine-tuning and transfer learning are techniques used to improve the performance of a model when insufficient data are available for the targeted task. Fine-tuning involves taking a pretrained model and training it on the target dataset using a small learning rate, in order to allow it to adapt to the target problem. In our particular scenario, the model was pretrained on an identical problem using a larger dataset from a similar domain.

Transfer learning is also aimed at improving the performance on target domains [31], but the pretrained model used as a starting point can be trained on a different problem, using data from a different domain. Part of the pretrained model is replaced, usually the last layers, in order to adapt it to the new problem. The resulting model is trained for a few epochs by updating only the new layers, and afterwards it can be trained in its entirety using a small learning rate.

## 4. Results

Our experiments targeted four different scenarios. First, we assessed the degree to which the selected machine learning models improve upon the baseline on the ULPC dataset. Second, we analyzed the performance of models trained on the ULPC dataset and tested on the children dataset in order to observe their capability to generalize out-of-the-box. Third, the SN and BERT-based models pretrained on the ULPC dataset were also fine-tuned on the children dataset. Last, we trained the models on a generic paraphrase identification dataset (i.e., MSRP) and adapted them to the paraphrase quality assessment via transfer learning. The open-source code used for these experiments can be found at [32].

The MSRP and ULPC datasets had a predetermined training-validation split which was maintained; for the children dataset, a random split was chosen and kept fixed throughout the experiments for consistency. Weighted F1-scores are reported to evaluate the performance of each model.

When considering the feature engineering approach, only the results obtained with the Extra Trees (ET) model are presented, as these models consistently surpassed all other configurations. Because these types of methods are not suited for transfer learning and fine-tuning, results for the ET model are available only in the first two Results subsections.

Several training runs of 20 epochs were conducted for the SN model, slightly varying the learning rates and the gamma by which the learning rate decreased each epoch. In the end, the best result was obtained using an Adam optimizer [33] and a starting learning rate of $1 \times 10^{-2}$, which was decreased by a factor of 0.95 every epoch, for 40 epochs. A weight decay factor of $1 \times 10^{-4}$ and a dropout of 0.1 were used for regularization.

Different strategies were tested for BERT in terms of fine-tuning, including the freeze of all BERT layers apart from the last, the usage of different learning rates, changing the pretrained BERT models (i.e., larger or smaller models, trained on cased or uncased texts). In the end a small version of BERT pretrained on cased texts was selected. The BERT-related parameters were trained using a small $1 \times 10^{-5}$ starting learning rate, while for the other parameters a $2 \times 10^{-2}$ learning rate was used. A scheduler that would linearly increase the learning rate in the first epochs, then decrease them exponentially using a gamma of 0.9 was used, along with and ADAMW optimizer [34]. For regularization a weight decay factor of $1 \times 10^{-4}$ was used for all parameters except biases and layer normalization [35] parameters. The experiments were run for 20 epochs.

### 4.1. ULPC Baseline

The original paper regarding the ULPC dataset contained the F1-scores for both low (scores between 1 and 3) and high (scores between 4 and 6) classes obtained using multiple indices as predictors. The initial set of indices included [21]: Latent Semantic Analysis (LSA) [36], overlap-indices, entailer indices (i.e., evaluating word and structure similarity via graph subsumption), minimal edit distances, as well as more rudimentary indices (e.g., length of response or difference in length between target sentence and response).

For each of the four dimensions, the baseline was selected as the index with the highest weighted F1 average. In the case of the Paraphrase Quality dimension, there were no results for the tripartite evaluation; as such, binary separation into low and high was chosen.

As it can be observed in Table 3, all three models obtained a better performance than the baseline, with an improvement ranging from 0.10 to 0.18 F1-score and with rather small differences in-between our models. The SN model achieved the best results for predicting semantic similarity, while the BERT-based model achieved the best results for the other three dimensions. Semantic similarity registered the highest improvement, bringing it closer in terms of performance to syntactic and lexical similarities. The results for the paraphrase quality dimension improved by 0.10 F1-score, reaching a 0.684 F1-score. Nevertheless, this dimension achieves an F1-score lower by 0.11 than the results for any of the other three dimensions, indicative of the challenges in assessing the overall quality of paraphrases.

**Table 3.** Performance of the models on the ULPC dataset.

| Model | Lexical Similarity (Avg F1) | Syntactic Similarity (Avg F1) | Semantic Similarity (Avg F1) | Paraphrase Quality (Avg F1) |
|---|---|---|---|---|
| Baseline | 0.716 | 0.707 | 0.611 | 0.583 |
| ET | 0.815 | 0.834 | 0.768 | 0.657 |
| SN | 0.795 | 0.839 | **0.794** | 0.658 |
| BERT | **0.816** | **0.860** | 0.791 | **0.684** |

*4.2. Out-of-the-Box Generalization on the Children Dataset*

The second set of experiments involved testing the accuracy of models trained on the ULPC dataset, directly on the children dataset. For this purpose, the ET, SN, and BERT models were trained on the entire ULPC dataset and tested on the children dataset on all four dimensions. A tripartite split into low (1–2), mid (3–4), and high (5–6) was used for the quality dimension, whereas the other dimensions were approached as binary classifications. As such, the results in Tables 4 and 5 were split—Table 4 covers the results for binary classification, while Table 5 targets a 3-class split. The split mentioned before leads to unbalanced classes (e.g., 31 samples in the 'Low' class compared to 84 samples in the 'High' class for the Lexical similarity dimension), but the performance of the models is good. Results argue that the ET model is the best model for out-of-the-box generalization, obtaining the highest average F1 score for all 4 dimensions. When comparing the two deep learning approaches, we observe that the BERT-based model manages to obtain better results on all tasks.

**Table 4.** Performance of ULPC-trained models on the children dataset for lexical, syntactic, and semantic similarity.

| Dimension | Model | Support (Low/High) | Low F1 | High F1 | Avg F1 |
|---|---|---|---|---|---|
| Lexical | ET | (31/84) | 0.806 | 0.929 | **0.895** |
| Lexical | SN | (31/84) | 0.422 | 0.629 | 0.573 |
| Lexical | BERT | (31/84) | 0.689 | 0.811 | 0.778 |
| Syntactic | ET | (35/80) | 0.688 | 0.776 | **0.749** |
| Syntactic | SN | (35/80) | 0.444 | 0.327 | 0.362 |
| Syntactic | BERT | (35/80) | 0.530 | 0.367 | 0.416 |
| Semantic | ET | (22/93) | 0.706 | 0.916 | **0.875** |
| Semantic | SN | (22/93) | 0.371 | 0.725 | 0.657 |
| Semantic | BERT | (22/93) | 0.575 | 0.802 | 0.758 |

**Table 5.** Performance of ULPC-trained models on the children dataset for overall paraphrase quality.

| Model | Support (Low/Mid/High) | Low F1 | Mid F1 | High F1 | Avg F1 |
|-------|------------------------|--------|--------|---------|--------|
| ET | (24/60/31) | 0.610 | 0.708 | 0.244 | **0.562** |
| SN | (24/60/31) | 0.333 | 0.337 | 0.205 | 0.300 |
| BERT | (24/60/31) | 0.454 | 0.712 | 0 | 0.466 |

When analyzing the results for paraphrase quality, we observe that all three models obtain poor results for the 'High' class indicative that a pair of sentences is a good paraphrase. The underlying reason might be a differentiated definition of a good paraphrase between the two datasets, given that they where generated based on inputs from users belonging to different age groups.

Despite obtaining the best results in this section, the ET model is not featured in the next experiments as it does not support fine-tuning or transfer learning. It is a lightweight model that can be trained easily and swiftly, and it does not overfit as easily as the deep learning models. Its main problem is the need for extensive feature engineering and feature selection prior to the start of the training process.

### 4.3. Fine-Tuning on the Children Dataset

In contrast to the previous experiment in which performance was assessed on the entire children dataset, now only its test partition was selected. This test set was kept the same across all fine-tuning and transfer learning experiments. The BERT-based and SN models were pretrained on the entire ULPC dataset and trained for a small number of epochs, using a smaller learning rate on the training subset of the children dataset. A learning rate of $1 \times 10^{-3}$ was used for the SN model, for a duration of 20 epochs. For the BERT model the same learning rate settings were used, but the number of epochs was decreased to 10. A weight decay factor of $1 \times 10^{-4}$, as well as a dropout factor of 0.1, were used for regularization. The same settings were used in the transfer learning sections as well.

The BERT-based model obtains higher results for the binary classification tasks (i.e., lexical, syntactic, and semantic, similarities from Table 6), while results for the more complex paraphrase quality task (see Table 7) are close to one another. Both models manage to improve by at least 0.14 average F1 score in all but one scenario, indicating the usefulness of fine-tuning in adapting the models to new datasets. The outlier is the BERT-based model for paraphrase quality, which achieved an increase of only 0.023 as the model is still incapable of identifying high-scored paraphrases in the children dataset.

**Table 6.** Performance of ULPC-trained models on the children dataset after fine tuning.

| Dimension | Model | Support (Low/High) | Pretrained Avg F1 (without Fine-Tuning) | Low F1 | High F1 | Avg F1 (with Fine-Tuning) |
|-----------|-------|--------------------|-----------------------------------------|--------|---------|---------------------------|
| Lexical | SN | (14/28) | 0.527 | 0.592 | 0.807 | 0.735 |
| Lexical | BERT | (14/28) | 0.767 | 0.965 | 0.981 | **0.976** |
| Syntactic | SN | (14/28) | 0.545 | 0.750 | 0.900 | 0.849 |
| Syntactic | BERT | (14/28) | 0.671 | 0.896 | 0.945 | **0.929** |
| Semantic | SN | (12/30) | 0.444 | 0.352 | 0.895 | 0.697 |
| Semantic | BERT | (12/30) | 0.656 | 0.600 | 0.875 | **0.796** |

**Table 7.** Performance of ULPC-trained models on the children dataset after fine-tuning (Paraphrase quality).

| Model | Support (Low/Mid/High) | Pretrained Avg F1 | Low F1 | Mid F1 | High F1 | Avg F1 |
|---|---|---|---|---|---|---|
| SN | (12/27/3) | 0.318 | 0.615 | 0.695 | 0.166 | **0.634** |
| BERT | (12/27/3) | 0.607 | 0.333 | 0.833 | 0 | 0.630 |

We once again observe poor performance for the high class in terms of paraphrase quality (see Table 8), underlining that criteria in-between the two datasets were different. Another potential explanation resides in the low number of examples for each class, which in turn may result in bias generated by particularly difficult outliers. However, the results from this experiment are consistent with the findings from the previous section, where high class examples were more abundant.

**Table 8.** Precision and Recall for the ULPC-trained models tested on the children dataset after fine-tuning (Paraphrase quality).

| Model | Support (Low/Mid/High) | Precision Low | Recall Low | Precision Mid | Recall Mid | Precision High | Recall High |
|---|---|---|---|---|---|---|---|
| SN | (12/27/3) | 0.571 | 0.667 | 0.842 | 0.593 | 0.111 | 0.333 |
| BERT | (12/27/3) | 0.278 | 0.417 | 0.952 | 0.741 | 0 | 0 |

### 4.4. Transfer Learning

The transfer learning task assesses the extent to which SN and BERT-based models trained for one task on a generic dataset can be successfully adapted to multiple tasks on another dataset. In this setup, both models were pretrained for binary paraphrase identification on either the MSRP dataset or the ULPC dataset, with paraphrase quality modeled as a binary classification. Afterwards, the last two layers were replaced with two random layers built for the target task (binary or 3-class classification), and the ensemble was trained for several epochs using a smaller learning rate. Multiple combinations of learning rate values and schedulers were used in order to ensure that the networks would not experience a sudden drop in performance (i.e., indicating that previously learned patterns were erased) and that the performance on the new dataset was improved upon.

We notice that the best results from Tables 9 and 10 are consistently obtained by the BERT model, and are better for models pretrained on the larger and more generic MSRP dataset than on the smaller, but more similar to the target, ULPC dataset. When comparing the results to the fine-tuning experiments, the best transfer learning model obtains better performance for semantic similarity (0.906 vs. 0.796) and paraphrase quality (0.694 vs. 0.634), identical performance for lexical similarity (0.976), and slightly poorer results for syntactic similarity (0.906 vs. 0.929). This means that in 3 out of 4 tasks (including the more complex paraphrase quality task), transfer learning on a more generic dataset yields comparable or better results than fine-tuning on a similar task.

Moreover, we observe that the results in Table 11 obtained for the Paraphrase Quality task are still poor when considering the high class, further underlining our previous observations. The assessment of whether a pair of sentences constitutes a good quality paraphrase is a difficult task, with definitions varying across datasets.

**Table 9.** Performance of ULPC/MSRP-trained models on the children dataset with transfer learning tuning.

| Dimension | Model | Support (Low/High) | Low F1 | High F1 | Avg F1 |
|---|---|---|---|---|---|
| **ULPC pretrained models** | | | | | |
| Lexical | SN | (14/28) | 0.480 | 0.779 | 0.679 |
| Lexical | BERT | (14/28) | 0.800 | 0.888 | **0.859** |
| Syntactic | SN | (14/28) | 0.583 | 0.833 | 0.750 |
| Syntactic | BERT | (14/28) | 0.866 | 0.925 | **0.906** |
| Semantic | SN | (12/30) | 0.400 | 0.812 | 0.694 |
| Semantic | BERT | (12/30) | 0.666 | 0.866 | **0.809** |
| **MSRP pretrained models** | | | | | |
| Lexical | SN | (14/28) | 0.210 | 0.769 | 0.582 |
| Lexical | BERT | (14/28) | 0.965 | 0.981 | **0.976** |
| Syntactic | SN | (14/28) | 0.416 | 0.766 | 0.649 |
| Syntactic | BERT | (14/28) | 0.866 | 0.925 | **0.906** |
| Semantic | SN | (12/30) | 0.352 | 0.835 | 0.697 |
| Semantic | BERT | (12/30) | 0.846 | 0.931 | **0.906** |

**Table 10.** Performance of ULPC/MSRP-trained models on the children dataset with transfer learning (Paraphrase quality).

| Model | Support (Low/Mid/High) | Low F1 | Mid F1 | High F1 | Avg F1 |
|---|---|---|---|---|---|
| **ULPC pretrained models** | | | | | |
| SN | (12/27/3) | 0.439 | 0.461 | 0 | 0.422 |
| BERT | (12/27/3) | 0.380 | 0.800 | 0.150 | **0.634** |
| **MSRP pretrained models** | | | | | |
| SN | (12/27/3) | 0.413 | 0.653 | 0 | 0.538 |
| BERT | (12/27/3) | 0.692 | 0.750 | 0.200 | **0.694** |

**Table 11.** Precision and recall for the ULPC/MSRP-trained models tested on the children dataset with transfer learning (Paraphrase quality).

| Model | Support (Low/Mid/High) | Precision Low | Recall Low | Precision Mid | Recall Mid | Precision High | Recall High |
|---|---|---|---|---|---|---|---|
| **ULPC pretrained models** | | | | | | | |
| SN | (12/27/3) | 0.310 | 0.750 | 0.750 | 0.333 | 0 | 0 |
| BERT | (12/27/3) | 0.444 | 0.333 | 0.870 | 0.741 | 0.100 | 0.333 |
| **MSRP pretrained models** | | | | | | | |
| SN | (12/27/3) | 0.353 | 0.500 | 0.727 | 0.593 | 0 | 0 |
| BERT | (12/27/3) | 0.643 | 0.750 | 0.857 | 0.667 | 0.143 | 0.333 |

In addition to the previous experiments, we assessed the performance of models undergoing a chain of transfer learning operations in two configurations: (a) generic-to-specific that simulates a gradual adaptation to a more specific dataset, and (b) specific-to-generic, which begins with a specific dataset, continues on the more generic dataset, and is fine-tuned on the final specific problem.

The models were trained on a first dataset, which was MSRP for "generic-to-specific" and ULPC for "specific-to-generic" for a binary paraphrase identification task. Then, they were retrained, using a smaller learning rate, on a second dataset (i.e., ULPC for "generic-to-specific" and MSRP for "Specific-to-generic") again for the binary paraphrase identification task. Afterwards, the same transfer learning as in the previous section was applied to adapt these models to the four paraphrase dimensions on the children dataset. This approach does require an extra stage of training, but the approach is still better than the fine-tuning one in terms of resource usage. A total of eight runs (four pretraining runs on ULPC and four fine-tuning runs on the children dataset) was required for training one model on the four dimensions of the children dataset using fine-tuning. In contrast, the same scenario using chained transfer learning requires six runs (one run on MSRP, one run on ULPC, and four runs on the children dataset).

Tables 12 and 13 depict that the "generic-to-specific" models outperform their counterparts in three out of four cases. Furthermore, we observe that results with chained transfer learning improve for the paraphrase quality and syntactic similarity tasks in comparison to the simple transfer learning approach, thus indicating the benefits of a chained transfer learning approach.

**Table 12.** Performance of models on the children dataset with transfer chained learning tuning.

| Dimension | Model | Support (Low/High) | Low F1 | High F1 | Avg F1 |
|---|---|---|---|---|---|
| **Generic-to-specific pretrained models** | | | | | |
| Lexical | SN | (14/28) | 0.300 | 0.781 | 0.62 |
| Lexical | BERT | (14/28) | 0.962 | 0.982 | **0.975** |
| Syntactic | SN | (14/28) | 0.480 | 0.779 | 0.679 |
| Syntactic | BERT | (14/28) | 0.896 | 0.945 | **0.929** |
| Semantic | SN | (12/30) | 0.526 | 0.861 | 0.765 |
| Semantic | BERT | (12/30) | 0.800 | 0.937 | **0.898** |
| **Specific-to-generic pretrained models** | | | | | |
| Lexical | SN | (14/28) | 0.500 | 0.843 | 0.729 |
| Lexical | BERT | (14/28) | 0.875 | 0.923 | **0.907** |
| Syntactic | SN | (14/28) | 0.600 | 0.875 | 0.783 |
| Syntactic | BERT | (14/28) | 0.875 | 0.923 | **0.907** |
| Semantic | SN | (12/30) | 0.526 | 0.861 | 0.765 |
| Semantic | BERT | (12/30) | 0.909 | 0.967 | **0.950** |

**Table 13.** Performance of models on the children dataset with chained transfer learning (Paraphrase quality).

| Model | Support (Low/Mid/High) | Low F1 | Mid F1 | High F1 | Avg F1 |
|---|---|---|---|---|---|
| **Generic -to-specific pretrained models** | | | | | |
| SN | (12/27/3) | 0.451 | 0.640 | 0 | 0.540 |
| BERT | (12/27/3) | 0.800 | 0.750 | 0.250 | **0.728** |
| **Specific-to-generic pretrained models** | | | | | |
| SN | (12/27/3) | 0.500 | 0.777 | 0.400 | 0.671 |
| BERT | (12/27/3) | 0.818 | 0.750 | 0.142 | **0.726** |

The performance of the models for the Paraphrase Quality dimension has improved (as seen in Tables 13 and 14) compared to previous scenarios. However, the results for the 'High' class are still modest. For the SN Specific-to-generic pretrained model we can observe better performance (higher precision and F1 score) compared to the equivalent BERT model, but the SN model's overall performance is still surpassed by its counterpart. This happens because the performance on the 'High' class is weighed down by the small number of samples.

**Table 14.** Precision and Recall for models tested on the children dataset with chained transfer learning (Paraphrase quality).

| Model | Support (Low/Mid/High) | Precision Low | Recall Low | Precision Mid | Recall Mid | Precision High | Recall High |
|---|---|---|---|---|---|---|---|
| **Generic-to-specific pretrained models** | | | | | | | |
| SN | (12/27/3) | 0.368 | 0.583 | 0.75 | 0.556 | 0 | 0 |
| BERT | (12/27/3) | 1 | 0.667 | 0.724 | 0.778 | 0.2 | 0.333 |
| **Specific-to-generic pretrained models** | | | | | | | |
| SN | (12/27/3) | 0.625 | 0.417 | 0.719 | 0.852 | 0.500 | 0.333 |
| BERT | (12/27/3) | 0.900 | 0.750 | 0.857 | 0.667 | 0.091 | 0.333 |

## 5. Discussion

In this work, we introduced three different models that managed to improve upon the baseline on the ULPC paraphrase quality assessment dataset across all dimensions: lexical, syntactic, semantic similarities, as well as paraphrase quality. The results were compared in

terms of a weighted F1 score across all classes, and all models improved upon the baseline result by at least 0.07 F1 score. In this scenario, the BERT model obtained the best result in three out of four cases, and was close to the best result, obtained by the SN model, in the fourth case. The ET model also obtained results close to the best ones across all four dimensions. The lowest average F1 scores were obtained for the paraphrase quality binary task, underlining that assessing paraphrase quality was the most difficult evaluation.

Afterwards, we verified the out-of-the-box generalization capabilities of the three models by taking models pretrained on the entire ULPC dataset and testing them on the entire children dataset. In this scenario, the best performing model was the ET model, which relied on extracting a wide range of features based on the input sentences, followed by a classical machine learning model trained on top of those features. This model easily surpassed the other models for all the four dimensions, indicating that the extracted features and the patterns found by the extra trees classifier were more general than the ones found by the deep learning models.

The third experiment was designed to observe if the performance of the deep learning models (SN and BERT) could be improved by fine tuning them for several epochs on the training subset of the children dataset and testing them on the validation subset. When comparing the results of the pretrained and fine-tuned networks, we noticed improvements of at least 0.14 average F1 score in all but one case. The fine-tuned BERT model obtained the best results by far, for the lexical, syntactic, and semantic similarity dimensions, and was a close second to the SN model for the paraphrase quality task.

The last experiment tested whether a model trained on a more generic dataset (MSRP or ULPC) and on a slightly different task (binary paraphrase detection) could be adapted to obtain good results when predicting the four dimensions on the children dataset. The results showed that in three out of four cases, the transfer learning approach obtained better results than fine-tuning. The results also showed that training on the larger and more generic MSRP dataset provided better results that training on the smaller, but more similar ULPC dataset. Further experiments also showed that chaining transfer learning operations can improve results for two of the four tasks, and the most efficient way of chaining transfer learning operations was from the most generic dataset to the most specific one.

## 6. Conclusions

In this work, we presented three different machine learning models achieving good results when predicting the following four paraphrase quality dimensions on the ULPC dataset: lexical, syntactic, semantic similarities, and paraphrase quality. The three models (ET, SN, and BERT) where chosen due to their different advantages and disadvantages, which were also evident throughout the experiments.

The ET model, which relies on feature engineering, required extensive efforts in extracting and filtering the features. Nevertheless, it obtained good initial results, and it had, by far, the best out-of-the-box performance when tested on the children dataset. Due to its setup, it was not appropriate for fine-tuning or transfer-learning scenarios. As such, it was only featured in the first experiments. The SN model, which is a deep learning model that has a medium number of parameters and can be trained from scratch, obtained good results in all the tasks, but it was outperformed by BERT-based models in terms of performance and by ET in terms of out-of-the-box generalization. The BERT-based model managed to consistently obtain the best or close to the best results. Tables 4 and 5 show one of the downsides of the BERT model, namely, that it can overfit on a small dataset and that it requires fine-tuning when required to solve an equivalent problem for a different dataset.

The fine-tuning and transfer learning approaches improved the performance for the SN and BERT models significantly. In most scenarios, the transfer learning approach obtained better results than the fine-tuning approach, despite the fact that the model used for transfer learning had been trained on a different problem (binary paraphrase identification), while the model used for fine-tuning had been trained on the same task as the target task. Small improvements were also noticed when training the models in a

sequence of transfer learning steps using the most generic dataset first, followed by more specific datasets; however, this approach requires more runtime for training.

When considering the four target paraphrase dimensions, there was a clear difference in terms of results between the three simple dimensions (i.e., lexical, syntactic, and semantic similarities) which were treated as a binary classification task, and the paraphrase quality dimension which was modeled as a 3-class classification problem. The difference between the simple dimensions and the overall dimension were of around 0.2 average F1-score for the most predictive models. When analyzing the three paraphrase quality classes, the models found the 'High' class to be the most difficult to predict, both for out-of-the-box generalization and for the fine-tuning/transfer learning tasks. We believe that this class might be more difficult to predict because the task of estimating whether a paraphrase is of good quality or not can depend on the dataset and domain. A good paraphrase for a dataset generated from news sources (MSRP) might not be considered a good paraphrase for a dataset constructed from rated human generated paraphrases (ULPC), which could receive lower scores for copy paste sequences of words, for instance. Similarly, the rigor applied to rating the paraphrases generated by adults (i.e., ULPC) might be higher than when rating children generated paraphrases (i.e., the children dataset). In addition, the small number of paraphrases rated as 'High' in terms of paraphrase quality can also explain these results, as outliers can significantly shift the results in their favor.

The conducted experiments suggest that the paraphrase assessment task consisting of four paraphrase dimensions can be tackled using both classical machine learning models and deep learning models that generalize well even on small datasets (~100 examples). Our experiments also show that for the deep learning models, the results can be improved by pretraining on larger and more generic datasets, such as by using a transfer learning approach to adapt the models to the smaller dataset.

In terms of future work, we aim to validate our models on other small paraphrase corpora from different domains in order to further verify their robustness. Recent advances in interpretability for deep learning models [37] can also be used to find the blind spots of the current models and improve their performance. Moreover, given the success of the chained transfer learning approach, we envision pretraining on even larger corpora, such as QQP, to further improve our models

**Author Contributions:** Conceptualization, D.S.M. and M.D.; methodology, M.D., B.N., and D.S.M.; software, B.N.; validation, B.N., M.D., and D.S.M.; formal analysis, M.D.; investigation, D.S.M.; resources, E.O.; data curation, N.N.N. and E.O.; writing—original draft preparation, B.N.; writing—review and editing, M.D., N.N.N., and D.S.M.; visualization, B.N.; supervision, D.S.M. and M.D.; project administration, D.S.M. and M.D.; funding acquisition, D.S.M. and M.D. All authors have read and agreed to the published version of the manuscript.

**Funding:** The work was funded by a grant of the Romanian National Authority for Scientific Research and Innovation, CNCS–UEFISCDI, project number TE 70 PN-III-P1-1.1-TE-2019-2209, ATES–"Automated Text Evaluation and Simplification" and by the Institute of Education Sciences (R305A190050) and the Office of Naval Research (N00014-17-1-2300 and N00014-20-1-2623). The opinions expressed are those of the authors and do not represent views of the IES or ONR.

**Institutional Review Board Statement:** The study was conducted according to the guidelines of the Declaration of Helsinki, and approved by the Institutional Review Board (or Ethics Committee) of Arizona State University (STUDY00010245 on 7-3-2019) as well as the Institutional Review Board of Hamline University (2020-07-103E on 7-3-2020). MSRP and ULPC are from external sources and are available freely online or upon request, respectively; both were developed by entities not related to the ones involved in this study.

**Informed Consent Statement:** Informed consent was obtained from all subjects involved in the study.

**Data Availability Statement:** The ULPC and MSRP datasets are available from external sources. The children dataset is private, but can be made available upon request.

**Conflicts of Interest:** The authors declare no conflict of interest.

## Abbreviations

The following abbreviations are used in this manuscript:

MSRP      Microsoft Research Paraphrase Corpus
ULPC      User Language Paraphrase Corpus
QQP       Quora Question Pair
RNN       Recurrent Neural Network
BERT      Bidirectional Encoder Representations from Transformers
BiLSTM    Bidirectional Long Short-Term Memory
NLP       Natural Language Processing
iSTART    Interactive Strategy Training for Active Reading and Thinking

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
