# Peer review of "Automated Paraphrase Quality Assessment Using Language Models and Transfer Learning"

_computers, doi:10.3390/computers10120166_

Round 1
Reviewer 1 Report
This article introduces a novel approach for automatically assessing the quality of paraphrases produced by individuals. Their approach is derived from the NLP task of paraphrase identification, which classifies a pair of texts as paraphrases of one another (or not). Their approach reduces the paraphrase quality task as a three-label classification problem, where the three classes are the previously mentioned ones plus a third one to identify a paraphrase pair of good quality. In order to address this task, the authors examine several models including a supervised classifier, Siamese networks based on BiLSTM RNNs, and pretrained BERT models, along with fine-tuning and transfer learning techniques. Both their BERT and Siamese network models seem to improve the state-of-the-art F1 scores. The authors also argue that their transfer learning technique is well suited for the task under consideration.
Overall, this is a well-written and well-structured paper, with nicely presented experimental results. Towards improving the quality of this work, I would like to point out the following issues:
- The authors consider the task of evaluating the quality of a paraphrase as a three-label classification problem. Have the authors considered a larger number of labels? Would it affect their approach? In one of the datasets, it is mentioned that for each sentence pair, a rating in the range [1, 6] is assigned. The authors split this range in 3 subranges in order to predict one out of three labels (not a paraphrase, paraphrase, paraphrase of good quality); one might argue that by adding more labels, their approach could capture more accurately the difference in quality between paraphrases.
- The pretrained models achieve really high accuracy after the transfer learning step for each dataset. What steps do the authors take in order to reduce overfitting? Please clarify.
- Authors need to explain in-depth the column “support (low / medium / high)” and its contents that appear in the tables with experimental results.
- In the Discussion section, the authors propose fine tuning the pre-trained models in a subset of the children paraphrases dataset which is already small (<1000 samples). This may introduce overfitting. If the smaller dataset is quite different from the larger one that was used to train the model, authors may want to consider using the weights of the intermediate layer of the pre-trained model as training input to a simple classifier, trained on this smaller dataset.
- Finally, to enhance the reproducibility of their work, I recommend authors to include a link to their employed datasets, as well as a link to the code of their approach and experiments.
Author Response
Response to Reviewer Comments
Dear Reviewer,
Thank you kindly for your feedback which helped us improve our manuscript!
Below we briefly explained the changes that were made.
Sincerely,
Bogdan Nicula
---
This article introduces a novel approach for automatically assessing the quality of paraphrases produced by individuals. Their approach is derived from the NLP task of paraphrase identification, which classifies a pair of texts as paraphrases of one another (or not). Their approach reduces the paraphrase quality task as a three-label classification problem, where the three classes are the previously mentioned ones plus a third one to identify a paraphrase pair of good quality. In order to address this task, the authors examine several models including a supervised classifier, Siamese networks based on BiLSTM RNNs, and pretrained BERT models, along with fine-tuning and transfer learning techniques. Both their BERT and Siamese network models seem to improve the state-of-the-art F1 scores. The authors also argue that their transfer learning technique is well suited for the task under consideration.
Overall, this is a well-written and well-structured paper, with nicely presented experimental results. Towards improving the quality of this work, I would like to point out the following issues:
- The authors consider the task of evaluating the quality of a paraphrase as a three-label classification problem. Have the authors considered a larger number of labels? Would it affect their approach? In one of the datasets, it is mentioned that for each sentence pair, a rating in the range [1, 6] is assigned. The authors split this range in 3 subranges in order to predict one out of three labels (not a paraphrase, paraphrase, paraphrase of good quality); one might argue that by adding more labels, their approach could capture more accurately the difference in quality between paraphrases.
Response: Thank you for the observation! We have considered using the full range of labels [1, 6], and the machine learning models that we used would have supported that scenario. However, we decided against this for 2 reasons. First, having to predict 1 out of 6 classes instead of 1 out of 2 or 3 increased the difficulty of the problem. Given the small size of the two paraphrase assessment datasets we chose a slightly easier task. Secondly, this existing split was the best common denominator for both the ULPC and children dataset. For the former the baseline was computed for this scenario and for the latter the scores already categorized (into 2 or 3 subranges) when we received it.
- The pretrained models achieve really high accuracy after the transfer learning step for each dataset. What steps do the authors take in order to reduce overfitting? Please clarify.
Response: We agree that given the size of the datasets some degree of overfitting was inevitable. To combat it we used weight decay and dropout and we monitored the training and validation loss. We have added extra commentary in order to better underline this.
- Authors need to explain in-depth the column “support (low / medium / high)” and its contents that appear in the tables with experimental results.
Response: The column illustrates the number of samples per class (low / medium / high). We tried to make this clearer by referencing those numbers in the text.
- In the Discussion section, the authors propose fine tuning the pre-trained models in a subset of the children paraphrases dataset which is already small (<1000 samples). This may introduce overfitting. If the smaller dataset is quite different from the larger one that was used to train the model, authors may want to consider using the weights of the intermediate layer of the pre-trained model as training input to a simple classifier, trained on this smaller dataset.
Response: We appreciate the suggestion, but we think that this scenario is partially covered by the transfer learning work. In that setting the last layers of the model are replaced with random parameters, and then the modified model is trained on the small dataset. This would be similar to adding a simple classifier (for instance an MLP model) on top of the pretrained network. The only difference is that we allow the pretrained parameters to continue learning using a conservative learning rate.
- Finally, to enhance the reproducibility of their work, I recommend authors to include a link to their employed datasets, as well as a link to the code of their approach and experiments.
Response: We have added a reference pointing to the repository containing the code that was used for the experiments. The ULPC dataset is available as part of the original paper referenced in out work. The children can not be made available at this moment.
Thank you for the overall feedback! We have tried to integrate the suggestions that were provided, and we think that this has improved the quality and readability of our manuscript.
Reviewer 2 Report
This paper evaluates three distinct approaches for paraphrase quality assessment: Extra Trees, Siamese Neural Networks, and a BERT-based approach.
Paraphrase identification and quality assessment is a relevant topic in the natural language processing domain and it can applied to several distinct tasks.
The comparative evaluation between the three referred approaches, was extended to allow an evaluation of the impact of fine-tuning and transfer learning methodologies.
The obtained results are interesting and partially support the paper conclusions.
However, I believe the paper would benefit if the following issues are better discussed/presented:
- F-measure is presented but there is no information about precision and recall. In this way it is not possible to know the main causes of errors: false positives or false negatives?
- Related with the previous point, there is no discussion about the main sources of errors: in which situations the classifiers are failing? Is it possible to organize errors in classes and discuss them? Some examples would also help (looking only to F-measure does not give any insight about the error situations and there is no discussion about this issue).
- It seems the best approach -- Extra Trees -- is dropped because it does not support fine tuning or transfer learning and requires feature engineering. But, for instance, ET could be trained with the ULPC+training set of children dataset and evaluated with the children testset, allowing a "soft" comparison with the fine tuning approach of the neural networks approach (I understand it is not the same concept but it would allow a "soft" comparison). Moreover, I believe the proposed features are dataset independent, so there is no need for additional efforts in the application of this methodology to a distinct dataset (only, maybe, in the feature reduction process).
- Regarding feature reduction, it is not clear how it was done and if the impact of the reduction was evaluated in the performance of the algorithms (namely, ET). A deeper discussion of this point should be done.
- In table 2 please explain/discuss the reason for some low values (PQ, for instance) and how this problem was handled.
- In page 7, please describe better how was the baseline calculated: "the baseline was select as the method with the highest weighted F1 average" -- a method from which set of methods? This issue was not clear to me.
Author Response
Response to Reviewer Comments
Dear Reviewer,
Thank you kindly for your feedback which helped us improve our manuscript!
Below we briefly explained the changes that were made.
Sincerely,
Bogdan Nicula
---
This paper evaluates three distinct approaches for paraphrase quality assessment: Extra Trees, Siamese Neural Networks, and a BERT-based approach.
Paraphrase identification and quality assessment is a relevant topic in the natural language processing domain, and it can be applied to several distinct tasks.
The comparative evaluation between the three referred approaches, was extended to allow an evaluation of the impact of fine-tuning and transfer learning methodologies.
The obtained results are interesting and partially support the paper conclusions.
However, I believe the paper would benefit if the following issues are better discussed/presented:
- F-measure is presented but there is no information about precision and recall. In this way it is not possible to know the main causes of errors: false positives or false negatives?
Response: Thank you for the observation! We added more information regarding precision and recall for the tasks that involved assessing the Paraphrase Quality dimension, to better understand the source of the failures.
- Related with the previous point, there is no discussion about the main sources of errors: in which situations the classifiers are failing? Is it possible to organize errors in classes and discuss them? Some examples would also help (looking only to F-measure does not give any insight about the error situations and there is no discussion about this issue).
Response: Thank you for pointing this out. We consider that the addition of the precision and recall scores as well as the more detailed discussion surrounding the low Kappa scores for the PQ dimension indicate some possible issues that the models faced. The fact that ‘low-effort’ paraphrases in which only one word was modified are assigned the ‘low’ class can confuse the models which mostly focus on the semantic resemblance between the 2 sentences. Also, the difference between how these models perform on the Semantic similarity dimension as opposed to the Paraphrase quality dimension indicates that these models are susceptible to scoring a jumbled or slightly altered version of the text as a ‘High’ class paraphrase.
- It seems the best approach -- Extra Trees -- is dropped because it does not support fine tuning or transfer learning and requires feature engineering. But, for instance, ET could be trained with the ULPC+training set of children dataset and evaluated with the children testset, allowing a "soft" comparison with the fine tuning approach of the neural networks approach (I understand it is not the same concept but it would allow a "soft" comparison). Moreover, I believe the proposed features are dataset independent, so there is no need for additional efforts in the application of this methodology to a distinct dataset (only, maybe, in the feature reduction process).
Response: Thank you for this observation! It is true that the Out-of-the-box generalization experiment indicates that the features on top of which the Extra Trees model is trained are suitable for both datasets. Training an Extra Trees model on a ULPC+children dataset would allow us to do a soft comparison between the 3 models in terms of fine-tuning. However, we see fine tuning as a precursor to transfer learning (which is more advantageous because we only need to train one model and adapt it to the 4 dimensions), for which Extra Trees model is also not well suited.
- Regarding feature reduction, it is not clear how it was done and if the impact of the reduction was evaluated in the performance of the algorithms (namely, ET). A deeper discussion of this point should be done.
Response: We added more details regarding the way in which the best features were selected at the end of section 3.2.1. After the filtering for constant and intercorrelated features was completed an ET model was trained and the top 100 most predictive features in it were selected to train a second ET model. The difference in performance between the 2 was negligible.
- In table 2 please explain/discuss the reason for some low values (PQ, for instance) and how this problem was handled.
Response: Thank you for underlining this. We added more details regarding the meaning of the low Kappa scores for the PQ dimensions in the paragraph preceding Table 2.
- In page 7, please describe better how was the baseline calculated: "the baseline was select as the method with the highest weighted F1 average" -- a method from which set of methods? This issue was not clear to me.
Response: We added more information regarding the way in which the baseline was computed in section 4.1.
Thank you for the overall feedback! We have tried to modify the manuscript according to your suggestions and we think that this has improved its quality.
Round 2
Reviewer 1 Report
I think authors have satisfactorily addressed my comments on the initial version.